# Dirac Cone Formation in Single-Component Molecular Conductors Based on Metal Dithiolene Complexes

**Reizo Kato** [1,*] and **Takao Tsumuraya** [2,3,*]

1    Condensed Molecular Materials Laboratory RIKEN, Saitama 351-0198, Japan
2    Priority Organization for lnnovation and Excellence (POIE), Kumamoto University, 2-39-1 Kurokami, Kumamoto 860-8555, Japan
3    Magnesium Research Center, Kumamoto University, 2-39-1 Kurokami, Kumamoto 860-8555, Japan
\*    Correspondence: reizo@riken.jp (R.K.); tsumu@kumamoto-u.ac.jp (T.T.)

**Abstract:** Single-component molecular conductors exhibit a strong connection to the Dirac electron system. The formation of Dirac cones in single-component molecular conductors relies on (1) the crossing of HOMO and LUMO bands and (2) the presence of nodes in the HOMO–LUMO couplings. In this study, we investigated the possibility of Dirac cone formation in two single-component molecular conductors derived from nickel complexes with extended tetrathiafulvalenedithiolate ligands, [Ni(tmdt)$_2$] and [Ni(btdt)$_2$], using tight-biding models and first-principles density-functional theory (DFT) calculations. The tight-binding model predicts the emergence of Dirac cones in both systems, which is associated with the stretcher bond type molecular arrangement. The DFT calculations also indicate the formation of Dirac cones in both systems. In the case of [Ni(btdt)$_2$], the DFT calculations, employing a vdW-DF2 functional, reveal the formation of Dirac cones near the Fermi level in the nonmagnetic state after structural optimization. Furthermore, the DFT calculations, by utilizing the range-separated hybrid functional, confirm the antiferromagnetic stability in [Ni(btdt)$_2$], as observed experimentally.

**Keywords:** Dirac electron systems; single-component molecular conductors; metal dithiolene complexes; tight-binding model; first-principles DFT calculation

## 1. Introduction

The conventional molecular conductors are categorized as multi-component systems wherein each molecule provides only one type of frontier molecular orbital (HOMO or LUMO) to form the conduction band, where HOMO and LUMO denote highest occupied molecular orbital and lowest unoccupied molecular orbital, respectively. The concept of a single-component molecular metal is based on a multi-orbital system in which more than two molecular orbitals (in this case, HOMO and LUMO) in the same molecule contribute to electronic properties. In a single-component molecular metal, the fully occupied HOMO band and the empty LUMO band overlap, and electron transfer between them induces a partially filled state. This idea was confirmed by the observation of electron and hole pockets in an ambient-pressure single-component molecular (semi)metal [Ni(tmdt)$_2$] (tmdt = trimethylenetetrathiafulvalenedithiolate: Scheme 1) through the detection of quantum oscillations at the begging of this century [1–4]. Since this breakthrough, various single-component molecular conductors have been developed by extending π conjugating systems to reduce the HOMO-LUMO energy gap, or by applying high pressure to increase the band width for each band. In particular, metal dithiolene complexes with a planar central core have played an important role due to their small HOMO-LUMO energy gap [5,6].

On the other hand, the Dirac electron system, typically observed in graphene, exhibits neither partially filled bands, like in metals, nor band gaps, like in insulators. The energy band structure of the Dirac electron system possesses a unique feature known as the Dirac

cone, where two conical surfaces with linear dispersion meet at a single point called the Dirac point in momentum space. A system in which the Dirac point forms extended lines or loops in the Brillouin zone is referred to as a nodal line semimetal, and has garnered significant attention due to the possibility of topologically nontrivial states [7,8].

The discovery of the nodal line semimetal state in a single-component molecular conductor [Pd(dddt)$_2$] (dddt = 5,6-dihydro-1,4-dithiin-2,3-dithiolate: Scheme 1) under high pressure has revealed a significant connection between single-component molecular conductors and the Dirac electron system [9]. The Dirac cone formation in [Pd(dddt)$_2$] can be understood using a tight-binding model based on extended Hückel molecular orbital calculations [10]. This mechanism relies on (1) the crossing of the HOMO and LUMO bands and (2) the presence of nodes in the HOMO–LUMO couplings, which favor the emergence of Dirac cones located near the Fermi level. Subsequently, an ambient-pressure nodal line semimetal based on a single-component molecular conductor [Pt(dmdt)$_2$] (dmdt = dimethyltetrathiafulvalenedithiolate: Scheme 1) was reported [11]. Our analysis using tight-binding band calculations indicated that this system exemplifies the mechanism proposed by us in a textbook manner [12]. An important aspect is the molecular arrangement (Figure 1) associated with the symmetry of the frontier molecular orbitals in [Pt(dmdt)$_2$], which satisfies the requirements for the Dirac cone formation. Within the *bc* plane, [Pt(dmdt)$_2$] molecules exhibit a stretcher bond pattern wherein each molecule overlaps with the molecule above and below it by half (Figure 1a). Transfer integrals labeled p and c are associated with major intermolecular interactions and form a two-dimensional conducting network (we tentatively call this network "layer"). Dirac cones are described on the $k_y$-$k_z$ plane, and nodal lines extend along the $k_x$ direction. As the HOMO has ungerade (odd) symmetry and the LUMO has gerade (even) symmetry (Figure 1b), the main HOMO–HOMO and LUMO–LUMO couplings (p and c) yield transfer integrals with opposite signs (Table 1), facilitating the crossing of the HOMO and LUMO bands, which is the first requirement for Dirac cone formation. In this work, we focus on two single-component molecular conductors derived from metal complexes with extended tetrathiafulvalenedithiolate ligands, [Ni(tmdt)$_2$] and [Ni(btdt)$_2$] (btdt = benzotetrathiafulvalenedithiolate: Scheme 1) [13], that exhibit molecular arrangements similar to that in [Pt(dmdt)$_2$]. The former is a (semi)metal, and the latter is a narrow-gap semiconductor with high conductivity at room temperature. We explore the possibility of Dirac cone formation in these compounds by means of tight-biding models and first-principles density functional theory (DFT) calculations.

[Pd(dddt)$_2$]

[Ni(tmdt)$_2$]

[Pt(dmdt)$_2$]

[Ni(btdt)$_2$]

**Scheme 1.** Component molecules of single-component molecular conductors.

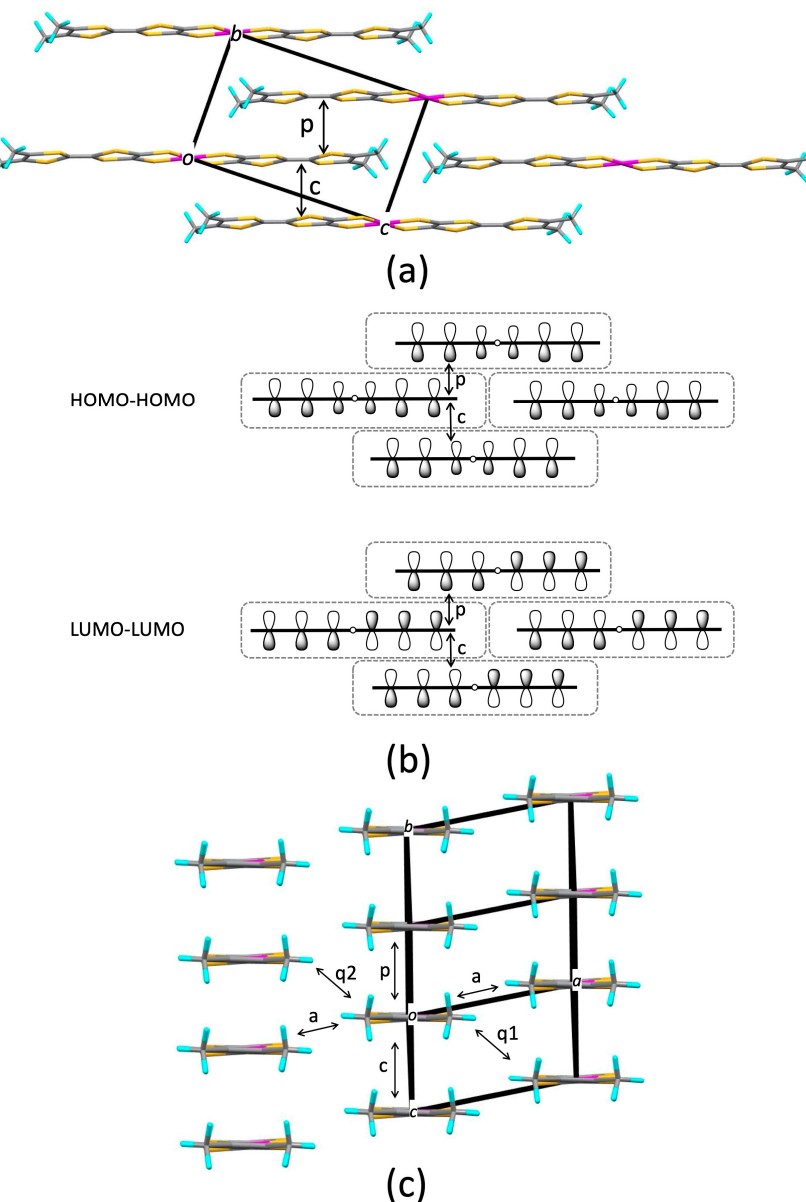

**Figure 1.** Crystal structure of [Pt(dmdt)$_2$]: (**a**) Molecular arrangement within the *bc* plane, (**b**) HOMO–HOMO and LUMO–LUMO couplings in the stretcher bond arrangement of the [Pt(dmdt)$_2$] molecule, where each frontier molecular orbital is represented using the p orbital of sulfur atom, (**c**) end-on projection (molecular arrangement viewed from the long molecular axis). Each element is identified by color as follows: S, yellow; C, gray; H, blue; Pt, magenta.

**Table 1.** HOMO–HOMO (H-H), LUMO–LUMO (L-L), and HOMO–LUMO (H-L) intermolecular transfer integrals (*t*) in [Pt(dmdt)$_2$] (meV). [1]

| Transfer Integral | H-H | L-L | H-L |
|:---:|:---:|:---:|:---:|
| $t_p$ | 53.4 | −49.8 | 51.7 |
| $t_c$ | 67.1 | −62.9 | 64.9 |
| $t_a$ | −6.2 | −6.5 | 0.3 |
| $t_{q1}$ | 8.2 | −7.4 | 7.8 |
| $t_{q2}$ | 8.2 | −7.7 | 7.9 |

[1] See Figure 1c. Transfer integrals in this table are used in Equations (2)–(4) with subscripts H-H, L-L, and H-L, which represent HOMO–HOMO, LUMO–LUMO, and HOMO–LUMO couplings, respectively (for example, $t_{pH\text{-}H}$ means a transfer integral $t_p$ between HOMO and HOMO).

## 2. Models and Methods

### 2.1. Tight-Binding Model

We used published atomic coordinates data [1,13] for our calculations. In order to make a comparison with [Pt(dmdt)$_2$] easier, each unit cell ($a_o$, $b_o$, $c_o$) was transformed to a new cell ($a$, $b$, $c$), as follows:

$a = a_o$, $b = b_o$, $c = a_o + c_o$ for [Ni(tmdt)$_2$]

$a = a_o$, $b = a_o + b_o$, $c = c_o$ for [Ni(btdt)$_2$].

In the tight-binding model, we consider one HOMO band and one LUMO band, because the unit cell contains only one molecule. Calculations of molecular orbitals and intermolecular overlap integrals ($S$) between HOMOs and LUMOs undertaken via the extended Hückel method were carried out using reported sets of semi-empirical parameters for Slater-type atomic orbitals and valence shell ionization potentials for H [14], C [14], Ni [14] and S [15]. Intermolecular transfer integrals, $t$ (eV), were estimated using the equation $t = -10\,S$. The band energies $E(k)$ ($k$ is given by $k = k_x a^* + k_y b^* + k_z c^*$ in terms of the reciprocal lattice vectors $a^*$, $b^*$, and $c^*$) are obtained as eigenvalues of the following $2 \times 2$ Hermitian matrix.

$$\mathbf{H}(k) = \begin{pmatrix} h_{\text{H-H}} & h_{\text{H-L}} \\ \overline{h_{\text{H-L}}} & h_{\text{L-L}} \end{pmatrix} \tag{1}$$

For [Pt(dmdt)$_2$], the matrix elements are given by

$$h_{\text{H-H}} = 2\,[t_{\text{pH-H}}\cos k(b+c) + t_{\text{cH-H}}\cos kc + t_{\text{aH-H}}\cos ka + t_{\text{q1H-H}}\cos k(a+c) + t_{\text{q2H-H}}\cos k(-a+b+c)] \tag{2}$$

$$h_{\text{H-L}} = 2i\,[t_{\text{pH-L}}\sin k(b+c) + t_{\text{cH-L}}\sin kc + t_{\text{aH-L}}\sin ka + t_{\text{q1H-L}}\sin k(a+c) + t_{\text{q2H-L}}\sin k(-a+b+c)] \tag{3}$$

$$h_{\text{L-L}} = \Delta + 2\,[t_{\text{pL-L}}\cos k(b+c) + t_{\text{cL-L}}\cos kc + t_{\text{aL-L}}\cos ka + t_{\text{q1L-L}}\cos k(a+c) + t_{\text{q2L-L}}\cos k(-a+b+c)], \tag{4}$$

where $\Delta$ is an energy gap between HOMO and LUMO (0.25 eV for [Pt(dmdt)$_2$]). Matrix elements $h_{\text{H-H}}$, $h_{\text{L-L}}$, and $h_{\text{H-L}}$ are associated with HOMO–HOMO, LUMO–LUMO, and HOMO–LUMO couplings, respectively.

### 2.2. DFT Calculations

We utilized various methods in our first-principles DFT calculations. To plot the Dirac cones in [Ni(tmdt)$_2$] and [Ni(btdt)$_2$], we performed band structure calculations using an all-electron full-potential linearized plane wave (FLAPW) method implemented in QMD-FLAPW12 [16,17]. The exchange-correlation functional employed was the generalized gradient approximation by Perdew, Burke, and Ernzerhof (GGA-PBE) [18]. We have found that a cut-off energy for plane waves of 20 Ry is sufficient for calculating the band structure. The cut-off energy for electron densities is assumed to be 213 Ry. For [Ni(tmdt)$_2$] and [Ni(btdt)$_2$], we employed uniform $k$-point meshes of $6 \times 6 \times 4$ and $8 \times 8 \times 4$, respectively. Since the $c$ axis direction is longer than the $a$ and $b$ directions in these crystal structures, the reciprocal lattice vector is shorter, allowing for a reduced number of $k$-point meshes along the $c$ direction. To calculate three-dimensional band dispersion of the Dirac cone, we used a high-density $k$-mesh, similar to our previous works [19,20].

For [Ni(btdt)$_2$] at ambient pressure, we performed structural relaxation using the van der Waals density functional (vdW-DF2) [21–23]. Specifically, we chose the vdW-DF2-b86r functional proposed by Hamada [24]. The calculations were carried out using the Quantum Espresso v.6.8 [25]. We employed the ultrasoft pseudopotential method with plane-wave basis sets. Ultrasoft pseudopotentials established by Garrity, Bennett, Rabe, and Vanderbilt (GBRV) were used [26]. During the structural relaxations, a uniform $k$-point mesh was set as $4 \times 4 \times 2$. When performing structural optimization with stress tensors, it is crucial to use a fixed number of plane waves based on the initial lattice parameters rather than a constant cut-off energy, as the latter can introduce significant errors in the calculated stress tensors [27]. To minimize errors in the calculated stress tensors, we used higher cut-off energies for plane waves, specifically, 60 Ry. The cut-off energy for electron densities was set to 488 Ry.

To investigate the stability of antiferromagnetic (AFM) ordering and the possible realization of the semiconducting phase in [Ni(btdt)$_2$], we employed a hybrid functional approach using the exchange-correlation functional set out by Heyd, Scuseria, and Ernzerhof (HSE06) [28,29]. The HSE06 calculations were performed by use of the Vienna Ab initio Simulation Package (VASP) [30–32], which is also based on the pseudopotential technique employing the projected augmented plane wave (PAW) method [33,34]. In the HSE06 calculations, we first obtained a converged charge density through the self-consistent calculations within GGA. Subsequently, self-consistent hybrid functional calculations were performed using the GGA charge density as the initial state. A *k*-point mesh of $3 \times 4 \times 2$ was employed for the calculations. The cut-off energy for plane waves was set to 36.75 Ry. The range-separation parameter in the HSE06 calculations was 0.2 Å$^{-1}$, and 25% of the exact exchange was mixed with the GGA exchange for short-range interactions.

## 3. Results

### 3.1. [Ni(tmdt)$_2$]

#### 3.1.1. Tight-Binding Model

Figure 2 illustrates the crystal structure of [Ni(tmdt)$_2$]. The molecular arrangement in the *bc* plane exhibits the stretcher bond pattern. The end-on projection indicates zigzag face-to-face stacking, and the molecular pair labeled p exhibits the strongest HOMO–HOMO and LUMO–LUMO couplings (Table 2). However, this does not imply a simple "dimerization", because the [Ni(tmdt)$_2$] molecules stack "uniformly" along the $b + c$ direction (corresponding to p) and along the $c$ direction (corresponding to c), as depicted in Figure 2a. Table 2 demonstrates that conducting layers parallel to the *bc* plane are strongly interconnected through interlayer transfer integrals q1 and a, which imports three-dimensional characteristics of the electronic structure. This differs from the case of [Pt(dmdt)$_2$] that fundamentally possesses a two-dimensional character. For most transfer integrals, including $t_\mathrm{p}$ and $t_\mathrm{c}$, the HOMO–HOMO and LUMO–LUMO couplings exhibit opposite signs.

**Table 2.** HOMO–HOMO (H-H), LUMO–LUMO (L-L), and HOMO–LUMO (H-L) intermolecular transfer integrals (*t*) in [Ni(tmdt)$_2$] (meV). [1]

| Transfer Integral | H-H | L-L | H-L |
|:---:|:---:|:---:|:---:|
| $t_\mathrm{p}$ | 72.0 | −68.8 | 70.5 |
| $t_\mathrm{c}$ | 33.9 | −26.7 | 30.1 |
| $t_\mathrm{a}$ | −32.8 | −33.1 | 0.6 |
| $t_\mathrm{q1}$ | 46.9 | −46.9 | 46.9 |
| $t_\mathrm{q2}$ | 3.6 | −3.2 | 3.4 |
| $t_\mathrm{r}$ | 2.8 | −2.6 | 2.7 |

[1] See Figure 2b. Transfer integrals in this table are used in Equations (5)–(7) with subscripts H-H, L-L, and H-L, that represent HOMO–HOMO, LUMO–LUMO, and HOMO–LUMO couplings, respectively.

The matrix elements of **H**(*k*) for [Ni(tmdt)$_2$] are given by

$$h_\text{H-H} = 2\left[t_\text{pH-H}\cos k(b+c) + t_\text{cH-H}\cos kc + t_\text{aH-H}\cos ka + t_\text{q1H-H}\cos k(c-a) + t_\text{q2H-H}\cos k(a+b+c) + t_\text{rH-H}\cos k(-a+b+c)\right] \quad (5)$$

$$h_\text{H-L} = 2i\left[t_\text{pH-L}\sin k(b+c) + t_\text{cH-L}\sin kc + t_\text{aH-L}\sin ka + t_\text{q1H-L}\sin k(c-a) + t_\text{q2H-L}\sin k(a+b+c)\right] + t_\text{rH-L}\sin k(-a+b+c)] \quad (6)$$

$$h_\text{L-L} = \Delta + 2\left[t_\text{pL-L}\cos k(b+c) + t_\text{cL-L}\cos kc + t_\text{aL-L}\cos ka + t_\text{q1L-L}\cos k(c-a) + t_\text{q2L-L}\cos k(a+b+c) + t_\text{rL-L}\cos k(-a+b+c)\right], \quad (7)$$

where $\Delta$ is 0.15 eV. Figures 3a and 4 display the band dispersion and Fermi surface obtained from the tight-binding model. On the $k_\mathrm{y}$-$k_\mathrm{z}$ plane ($k_\mathrm{x} = 0$), a crossing of the HOMO and LUMO bands occurs at $(k_\mathrm{x}, k_\mathrm{y}, k_\mathrm{z}) = (0.0, \pm 0.578, \pm 0.210)$ (marked as N in Figure 3a), resulting in the emergence of symmetrical Dirac cones around the Γ point (0.0, 0.0, 0.0) (Figure 5). The Dirac points reside below the Fermi level, resulting in electron pockets. Due to the strong interlayer interactions among the conducting networks parallel to the *bc* plane, the Dirac points move within the three-dimensional wave vector space depending on $k_\mathrm{x}$ and form a loop. The reciprocal unit cell contains one nodal loop, which is nearly

planar. The loop exhibits energy variations, which gives rise to electron pockets and hole pockets, indicating the presence of a nodal line semi-metal (Figure 6). Compared to [Pt(dmdt)$_2$], however, the deviation of the band energy from the Fermi energy is relatively large, resulting in a system with a large Fermi surface (Figure 4).

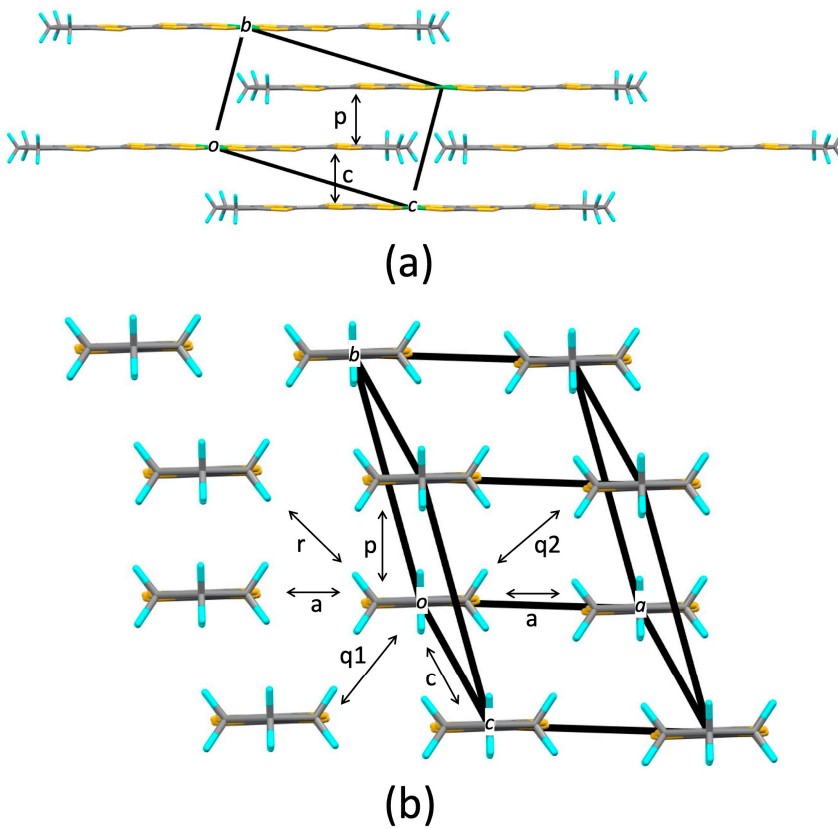

**(a)**

**(b)**

**Figure 2.** Crystal structure of [Ni(tmdt)$_2$]: (**a**) Molecular arrangement within the *bc* plane, (**b**) end-on projection. Each element is identified by color as follows: S, yellow; C, gray; H, blue; Ni, green.

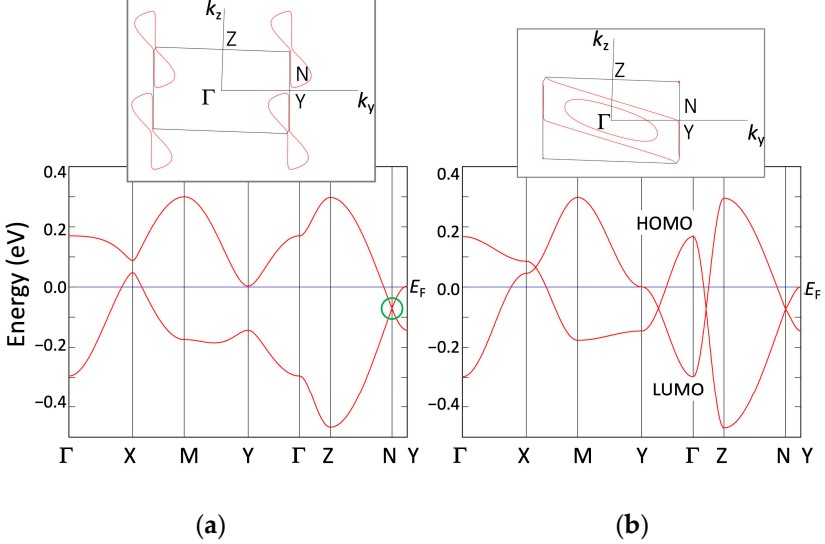

**(a)**  **(b)**

**Figure 3.** Band dispersion and Fermi surface (on the $k_x = 0$ plane) of [Ni(tmdt)$_2$] (**a**) with HOMO–LUMO couplings (pristine), (**b**) without HOMO–LUMO couplings ($h_{H-L} = 0$). Γ = (0,0,0), X = (1/2,0,0), M = (1/2,1/2,0), Y = (0,1/2,0), Z = (0,0,1/2), and N = (0,0.578,0.210), in units of the reciprocal lattice vectors. N denotes a position of the Dirac point.

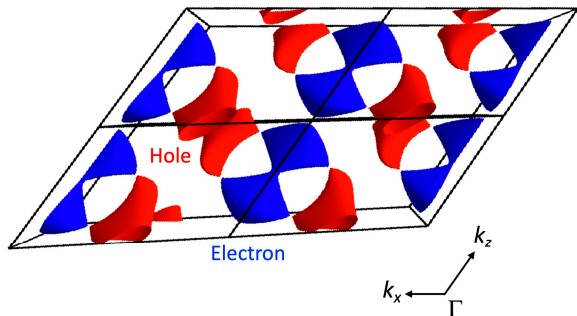

**Figure 4.** Fermi surface of [Ni(tmdt)$_2$] viewed from the $k_y$ axis. This figure is plotted with FermiSurfer version 2.1 [35].

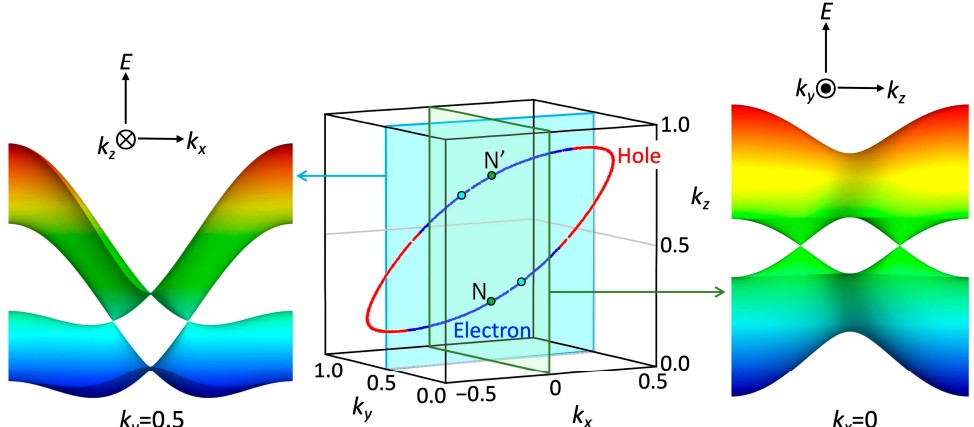

**Figure 5.** Loop formed by Dirac points (Nodal line) in the three-dimensional wave vector space and a pair of Dirac cones on the $k_x = 0$ and $k_y = 0.5$ planes in [Ni(tmdt)$_2$]. The hole-like characteristic is indicated in red and the electron-like characteristic in blue.

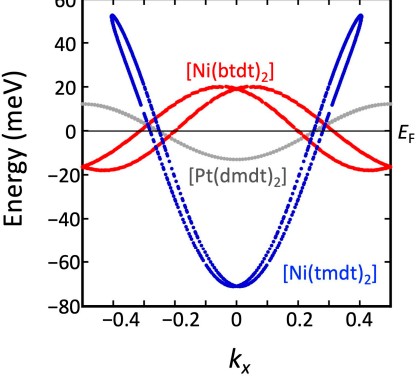

**Figure 6.** $k_x$ dependence of the band energy at the Dirac point calculated with tight-binding models.

### 3.1.2. First-Principles DFT Calculations

Figure 7 indicates the DFT band structure of [Ni(tmdt)$_2$] calculated with the GGA-PBE functional. In the $k_x = 0$ plane, due to the space inversion symmetry, a pair of Dirac points exists at $(k_x, k_y, k_z) = (0, 0.685, 0.265)$ and $(0, 0.315, -0.265)$. The Dirac point is located 65 meV lower than the Fermi level. The tight-binding band structure in Figure 3a is generally in agreement with the DFT band structure, e.g., hall-like characteristics near the X point and electron-like characteristics near the N point.

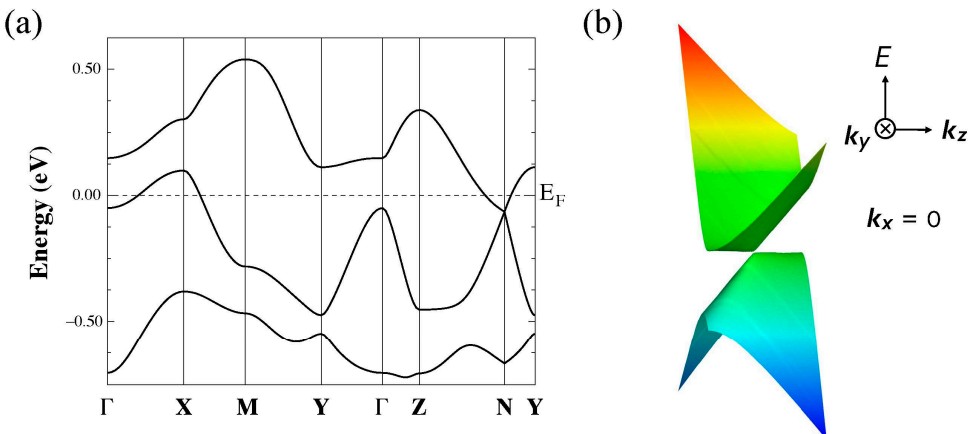

**Figure 7.** (**a**) Band structure of [Ni(tmdt)$_2$] calculated with GGA-PBE. Γ = (0,0,0), X = (1/2,0,0), M = (1/2,1/2,0), Y = (0,1/2,0), Z = (0,0,1/2), and N = (0,0.685,0.265). N represents a position of the Dirac point. (**b**) Dirac cone on the $k_x$ = 0 plane.

*3.2. [Ni(btdt)$_2$]*

3.2.1. Tight-Binding Model

As shown in Figure 8a, the molecular arrangement of [Ni(btdt)$_2$] within the *bc* plane exhibits the stretcher bond pattern. The interplanar distance in the molecular pair labeled c (4.24 Å) is significantly longer than that in the molecular pair p (3.01 Å), leading to a considerable reduction in intermolecular interaction c. In contrast, in [Pd(dmdt)$_2$] and [Ni(tmdt)$_2$], the corresponding interplanar distances are nearly equivalent (3.54 Å and 3.53 Å in [Pd(dmdt)$_2$], 3.46 Å and 3.58 Å in [Ni(tmdt)$_2$]). The main HOMO–HOMO and LUMO–LUMO couplings are observed in the molecular pairs labeled p and q1 (Table 3, Figure 8b). Both transfer integrals $t_p$ and $t_{q1}$ exhibit HOMO–HOMO and LUMO–LUMO couplings with opposite signs. Notably, $t_p$ is opposite in sign to $t_p$ in [Pt(dmdt)$_2$] and [Ni(tmdt)$_2$].

**Table 3.** HOMO–HOMO (H-H), LUMO–LUMO (L-L), and HOMO–LUMO (H-L) intermolecular transfer integrals (*t*) in [Ni(btdt)$_2$] (meV). [1]

| Transfer Integral | H-H | L-L | H-L |
|:---:|:---:|:---:|:---:|
| $t_p$ | −59.8 | 51.0 | −55.5 |
| $t_c$ | 6.6 | 2.5 | 1.7 |
| $t_a$ | 7.9 | 9.6 | 0.2 |
| $t_{q1}$ | 44.8 | −24.5 | 34.2 |
| $t_{q2}$ | 0.6 | −0.6 | 0.6 |
| $t_s$ | −1.7 | 1.5 | −1.6 |

[1] See Figure 8b. Transfer integrals in this table are used in Equations (8)–(10) with subscripts H-H, L-L, and H-L, that represent HOMO–HOMO, LUMO–LUMO, and HOMO–LUMO couplings, respectively.

The matrix elements of **H**(*k*) for [Ni(btdt)$_2$] are given by

$$h_{\text{H-H}} = 2\,[t_{\text{pH-H}}\cos k(b+c) + t_{\text{cH-H}}\cos kc + t_{\text{aH-H}}\cos ka + t_{\text{q1H-H}}\cos k(c+a) + t_{\text{q2H-H}}\cos k(-a+b+c) + t_{\text{sH-H}}\cos k(b+2c)] \quad (8)$$

$$h_{\text{H-L}} = 2i\,[t_{\text{pH-L}}\sin k(b+c) + t_{\text{cH-L}}\sin kc + t_{\text{aH-L}}\sin ka + t_{\text{q1H-L}}\sin k(c+a) + t_{\text{q2H-L}}\sin k(-a+b+c)] + t_{\text{sH-L}}\sin k(b+2c)] \quad (9)$$

$$h_{\text{L-L}} = \Delta + 2\,[t_{\text{pL-L}}\cos k(b+c) + t_{\text{cL-L}}\cos kc + t_{\text{aL-L}}\cos ka + t_{\text{q1L-L}}\cos k(c+a) + t_{\text{q2L-L}}\cos k(-a+b+c) + t_{\text{sL-L}}\cos k(b+2c)], \quad (10)$$

where Δ is 0.15 eV. The band dispersion and Fermi surface (Figures 9a and 10) obtained from the tight-binding model indicate a semimetallic electronic structure where hole and electron pockets align alternately along the *a*\* + *b*\* − *c*\* direction. The Dirac points emerge at ($k_x$, $k_y$, $k_z$) = (0.0, ±0.118, ±0.264) (marked as N in Figure 8a) on the $k_y$-$k_z$ plane ($k_x$ = 0) and (±0.308, ±0.404, 0.0) on the $k_x$-$k_y$ plane ($k_z$ = 0). In the three-dimensional reciprocal

lattice, the Dirac point forms a line along the $a^* + b^* - c^*$ direction (Figure 11). The energy at the Dirac point fluctuates around the Fermi level along the line (Figure 6), resulting in the presence of hole and electron pockets.

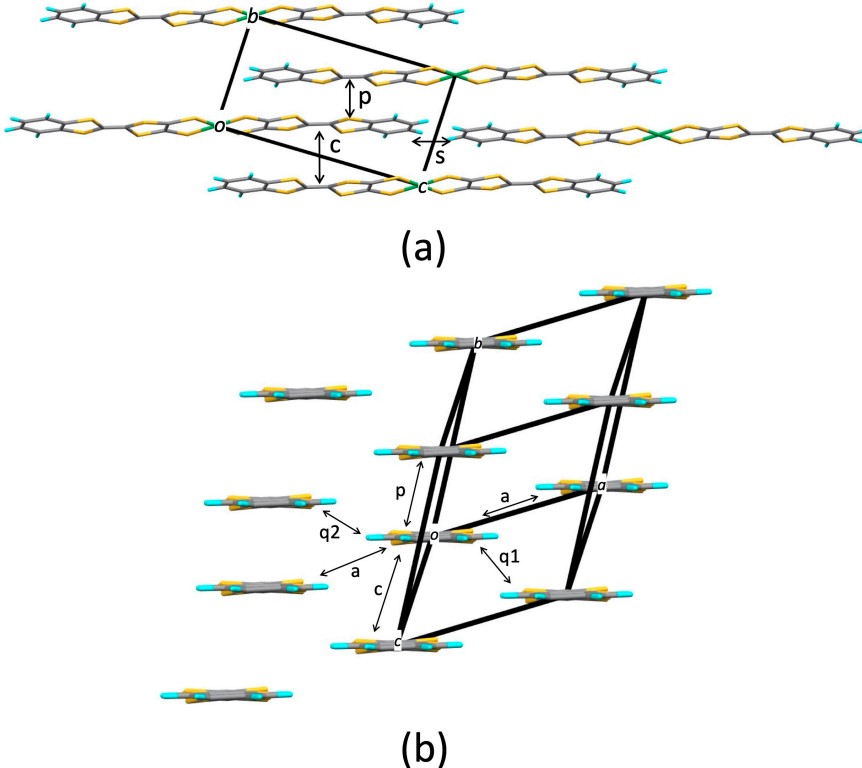

(a)

(b)

**Figure 8.** Crystal structure of [Ni(btdt)$_2$]: (**a**) Molecular arrangement within the *bc* plane, (**b**) end-on projection. Each element is identified by color as follows: S, yellow; C, gray; H, blue; Ni, green.

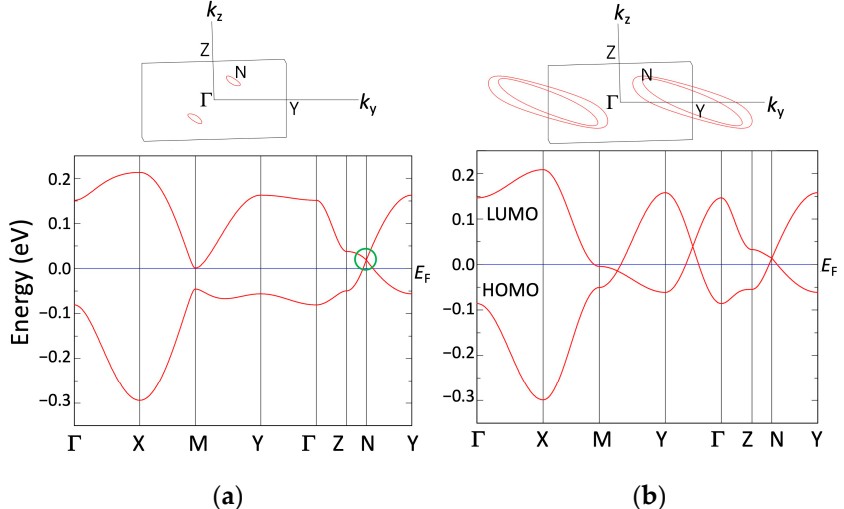

(**a**)          (**b**)

**Figure 9.** Band dispersion and Fermi surface (on the $k_x = 0$ plane) of [Ni(btdt)$_2$] (**a**) with HOMO–LUMO couplings (pristine), (**b**) without HOMO–LUMO couplings ($h_{\text{H-L}} = 0$). $\Gamma = (0,0,0)$, X $= (1/2,0,0)$, M $= (1/2,1/2,0)$, Y $= (0,1/2,0)$, Z $= (0,0,1/2)$, and N $= (0,0.118,0.264)$, in units of the reciprocal lattice vectors. N denotes a position of the Dirac point.

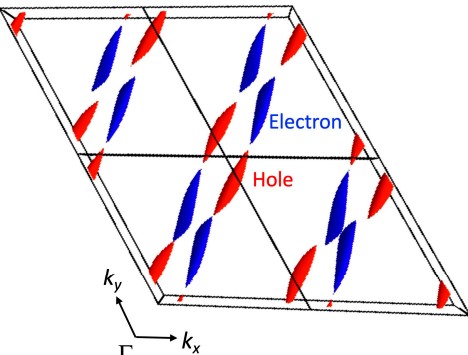

**Figure 10.** Fermi surface of [Ni(btdt)2] viewed from the $k_z$ axis. This figure is plotted with FermiSurfer version 2.1 [33].

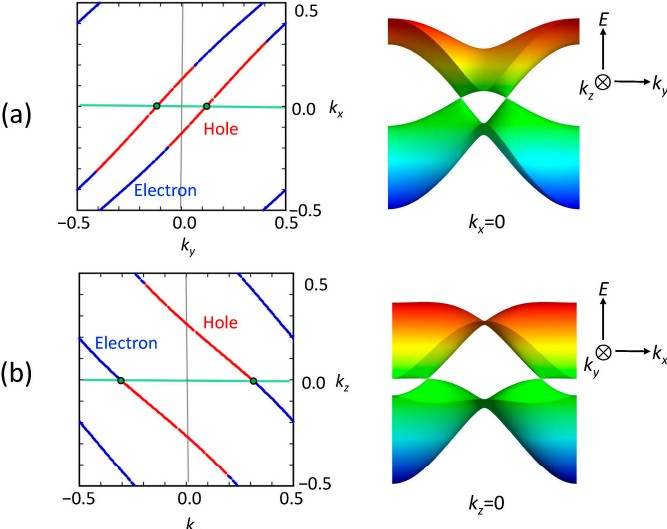

**Figure 11.** Nodal line projected on the $k_x$-$k_y$ (**a**) and $k_x$-$k_z$ (**b**) planes and a pair of Dirac cones on the $k_x = 0$ and $k_z = 0$ planes in [Ni(btdt)$_2$]. The hole-like characteristic is indicated in red and the electron-like characteristic in blue.

### 3.2.2. First-Principles DFT Calculations

Next, we describe the structural and electronic properties of [Ni(btdt)$_2$] as calculated using the first-principles method. Figure 12a shows the band structure for the experimental crystal structure with the original unit cell vectors, indicating a metallic character. Unlike the tight-binding model, the DFT band structure does not exhibit a Dirac cone.

As mentioned in the previous subsection, the experimental crystal structure, determined using powder X-ray diffraction analysis with constrained conditions, has a much shorter interplanar distance (~3.01 Å) compared to other single-component metal dithiolene complexes such as [Ni(tmdt)$_2$] [1] and [Ni(hfdt)$_2$] (hfdt = bis(trifluoromethyl) tetrathiafulvalenedithiolate) (~3.40 Å) [36]. Therefore, we performed structural relaxation for the lattice parameters and internal coordinates using a vdW-DF2 functional (vdw-df2-b86r) [28]. The vdW-DF method, proposed by Dion et al. [21], incorporates a semi-local exchange-correlation functional and a nonlocal correlation functional to account for dispersion interactions. In single-component molecular crystals, dispersion interactions are significant at ambient and low-pressure conditions. We anticipate that structural relaxation with the vdW-DF functional will yield an efficient determination of structural properties.

Figure 12b displays the band structure for the structure with relaxed atomic positions using the vdw-df2-b86r functional (Appendix A). The band gap remains unopened, and a massless Dirac cone emerges at the Fermi level. The band at the Y point has shifted lower, and there is a significant separation between the top and lower bands of the valence

band. A new unit cell described in Section 2.1 for the optimized structure has the lattice parameters $a$ = 6.36, $b$ = 6.98, $c$ = 14.46 Å, $\alpha$ = 107.53, $\beta$ = 78.34, and $\gamma$ = 79.80°, while those for the experimental structure are $a$ = 6.720, $b$ = 7.925, $c$ = 12.563 Å, $\alpha$ = 90.41, $\beta$ = 93.62, and $\gamma$ = 59.85°. The DFT-optimized structure, where the interplanar distances are adjusted to ordinary values (~3.47 Å), appears to be a reasonable structure. However, despite using the DFT-optimized structure to simulate the XRD pattern, the resulting simulated pattern differs from the pattern obtained through the powder X-ray diffraction method. Hence, we consider the DFT-optimized structure as a model structure.

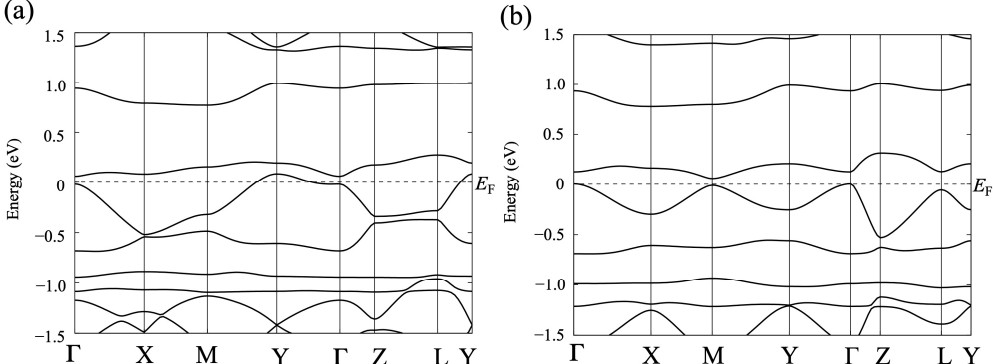

**Figure 12.** (**a**) Band structure for the experimental structure of [Ni(btdt)$_2$] calculated with GGA-PBE functional. (**b**) Band structure for the optimized structure calculated with a vdW-DF2 functional (vdw-df2-b86r). These band structures are calculated based on the original unit cell vectors. $\Gamma$ = (0,0,0), X = (1/2,0,0), M = (1/2,1/2,0), Y = (0,1/2,0), Z = (0,0,1/2), and L = (0,1/2,1/2), in units of the reciprocal lattice vectors.

Figure 13a illustrates the band structure for the new unit cell of the optimized structure. The Dirac points are located at specific points in **k**-space, represented by the coordinates $(k_x, k_y, k_z)$ = ($\pm$0.3120, $\mp$0.014, 0). Figure 13b and Figure 13c depict the Dirac cones observed in the $k_x$-$k_y$ and $k_x$-$k_z$ planes, respectively. It should be added that the tight-binding band calculation based on the DFT-optimized structure also indicates the presence of Dirac cones at $(k_x, k_y, k_z)$ = ($\pm$0.414, 0, $\mp$0.004).

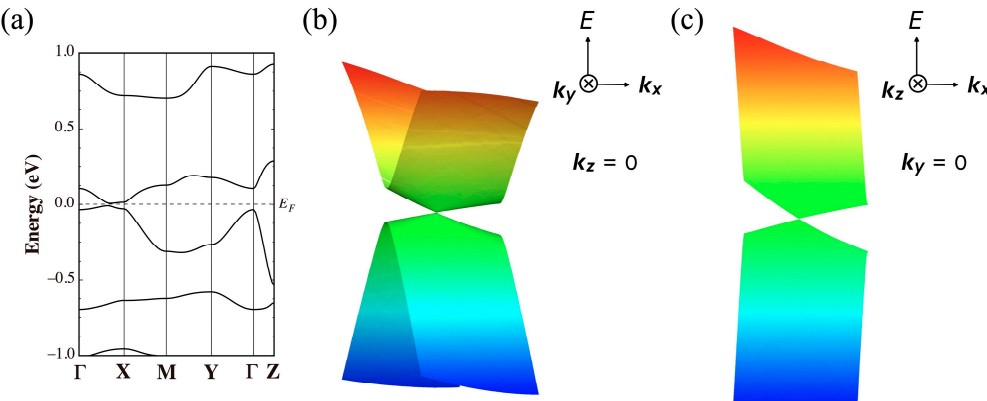

**Figure 13.** (**a**) Band dispersion of nonmagnetic calculations in [Ni(btdt)$_2$] for the crystal structure optimized with a vdW-DF2 functional (vdw-df2-b86r). $\Gamma$ = (0,0,0), X = (1/2,0,0), M = (1/2,1/2,0), Y = (0,1/2,0), Z = (0,0,1/2), and L = (0,1/2,1/2), for the new unit cell. Band structure close to the Dirac point on (**b**) the $k_x$-$k_y$ and (**c**) $k_x$-$k_z$ planes. These electronic structures are calculated with the GGA-PBE functional using the all-electron FLAPW method.

## 4. Discussion

### 4.1. Dirac Cone Formation in Single-Component Molecular Conductors

The tight-binding models have revealed that all three single-component molecular conductors with the stretcher bond type molecular arrangement, [Pd(dmdt)$_2$], [Ni(tmdt)$_2$], and [Ni(btdt)$_2$], form Dirac cones. The emergence of the Dirac cone is determined by satisfying the following conditions simultaneously in $\mathbf{H}(\mathbf{k})$ [12]:

$$h_{\text{H-H}} = h_{\text{L-L}} \tag{11}$$

$$h_{\text{H-L}} = 0. \tag{12}$$

Equation (11) represents the HOMO–LUMO band crossing, and Equation (12) indicates a node of the HOMO–LUMO coupling. When the HOMO–LUMO coupling $h_{\text{H-L}}$ is set to zero, the HOMO and LUMO bands intersect, resulting in a large Fermi surface as shown in Figures 3b and 9b. Notably, the energy at the intersection is situated near the Fermi level, because the HOMO is fully occupied and the LUMO is entirely empty in the isolated molecule state. Figure 14 illustrates that the intersection forms a (green) loop on the $k_y$-$k_z$ plane ($k_x = 0$) for [Ni(tmdt)$_2$] and [Ni(btdt)$_2$]. In real systems, however, the HOMO–LUMO coupling induces band hybridization and maintains a gap between the two bands, known as band repulsion. An exception occurs at the node of the HOMO–LUMO coupling. In Figure 14, the node ($h_{\text{H-L}} = 0$) is represented by the orange line, and the intersection point with the green loop gives the Dirac point where Equations (11) and (12) are satisfied.

This mechanism operates as long as the HOMO and LUMO bands cross each other. The band crossing arises due to the small energy gap between the HOMO and LUMO ($\Delta$), leading to the emergence of Dirac cones within a specific range of the $\Delta$ value ($\Delta < 0.62$ eV for [Ni(tmdt)$_2$] and $\Delta < 0.37$ eV for [Ni(btdt)$_2$], respectively). Furthermore, the curvature of the two bands is also a crucial factor for the band crossing. Bands with opposite curvatures tend to intersect each other. The band curvature is associated with the sign of the transfer integrals. The stretcher bond type molecular arrangement facilitates HOMO–HOMO and LUMO–LUMO transfer integrals with opposite signs. In Figure 15, the transfer integrals in the face-to-face stacking arrangement are mapped as a function of the offset from the center of the molecule (Pt or Ni) for the [Pt(dmdt)$_2$] and [Ni(tmdt)$_2$] molecules. The maps for both compounds are very similar to each other. In the region of Figure 15, there are two (positive or negative) peaks around $s = 0$ Å and three (positive or negative) peaks around $s = 1.75$ Å. The transfer integrals p are situated around $s = 0$ Å and the transfer integrals c are located around $s = 0$ Å ([Pt(dmdt)$_2$]) or 1.75 Å ([Ni(tmdt)$_2$]). Importantly, the HOMO–HOMO and LUMO–LUMO transfer integrals exhibit opposite signs almost everywhere.

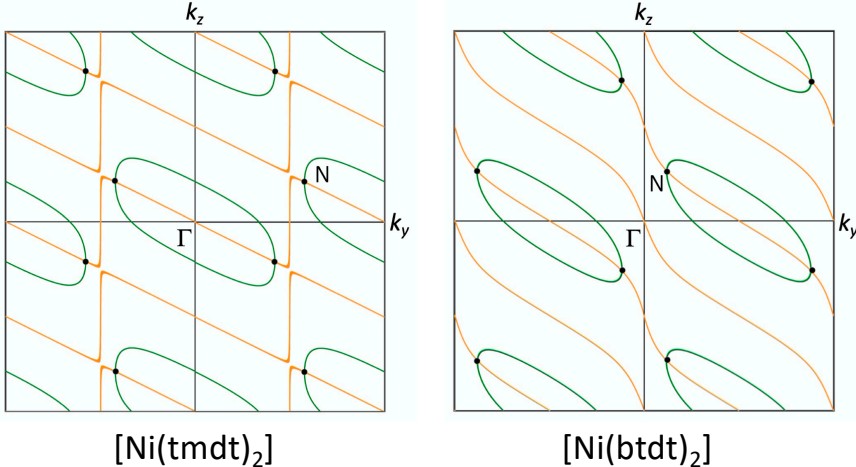

[Ni(tmdt)$_2$]  [Ni(btdt)$_2$]

**Figure 14.** Intersection of the HOMO and LUMO bands ($h_{\text{H-H}} = h_{\text{L-L}}$: green loop), node of the HOMO–LUMO coupling ($h_{\text{H-L}} = 0$: orange line), and the Dirac points (black dots) on the $k_x = 0$ plane (See text).

The [Ni(btdt)$_2$] molecule, with the terminal benzene rings, provides a different pattern in the maps within the $L > 14$ Å region (Figure 16). Nevertheless, even in this case, the HOMO–HOMO and LUMO–LUMO transfer integrals maintain opposite signs.

As observed above, although the face-to-face stacking mode in the stretcher bond type molecular arrangement displays several variations, each mode tends to yield significantly strong HOMO–HOMO and LUMO–LUMO couplings with opposite signs, thereby inducing band crossing. In the early stages of research on single-component molecular conductors, the presence of crossing band structures was considered undesirable for achieving a metallic state due to the band repulsion, and a parallel band structure composed of HOMO and LUMO bands with the same curvature was preferred [5]. This viewpoint, however, is applicable only to one-dimensional systems. Currently, single-component molecular metals with two- or three-dimensional crossing band structures are not uncommon and can be understood within the framework of the conventional tight-binding band picture. The truly unique aspect of single-component molecular conductors with crossing band structure is the emergence of the Dirac electron system, which represents a new direction for the study of functional single-component molecular conductors.

Unfortunately, both [Ni(tmdt)$_2$] and [Ni(btdt)$_2$] do not clearly exhibit the nature of the Dirac electron system. In [Ni(tmdt)$_2$], the (semi)metallic nature associated with the large Fermi surface dominates electronic properties. This is because the energy at the Dirac point is largely shifted away from the Fermi level (Figure 6). On the other hand, for [Ni(btdt)$_2$], we considered the effects of antiferromagnetic (AFM) spin order, as demonstrated in the following subsection.



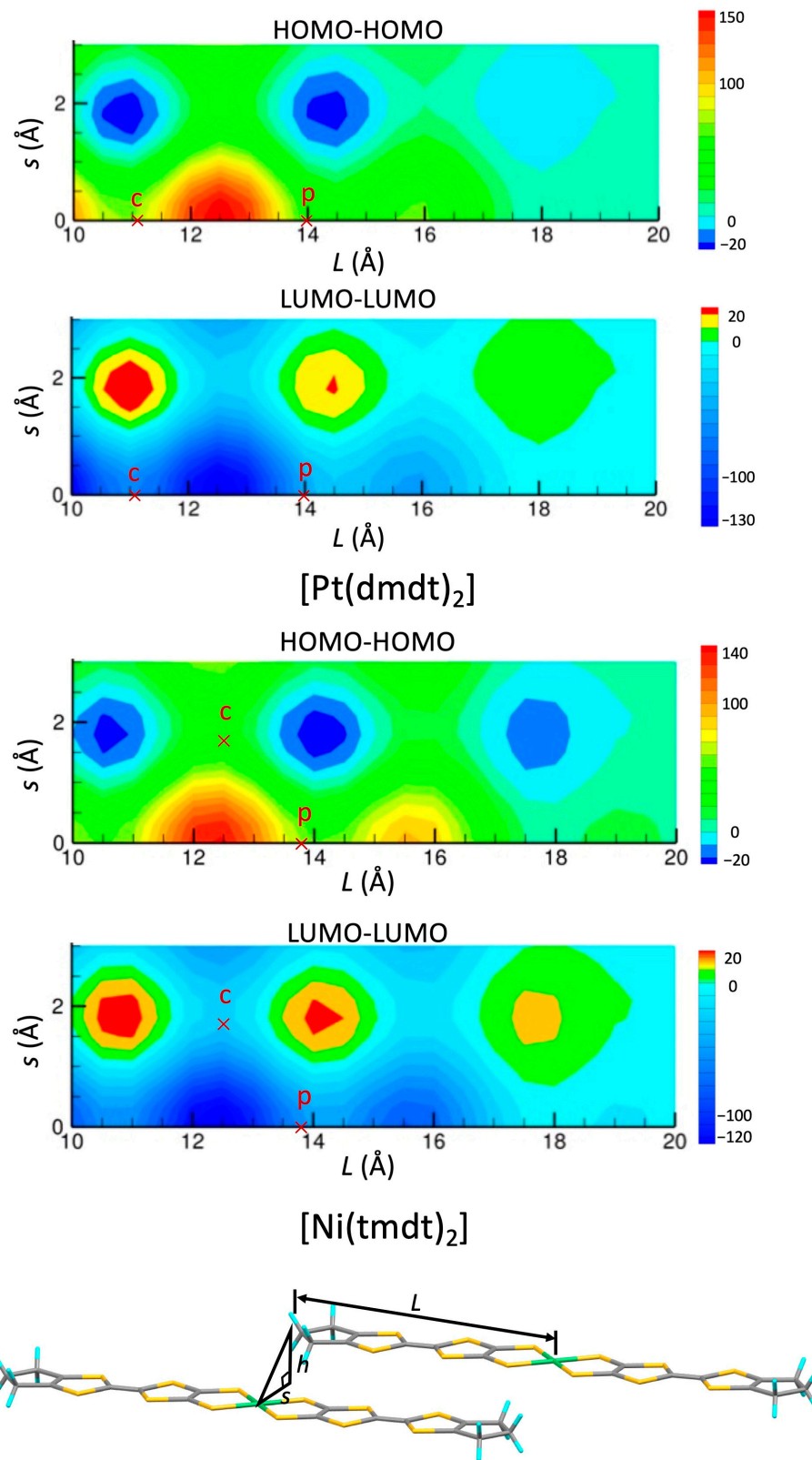

**Figure 15.** Map of HOMO–HOMO and LUMO–LUMO transfer integrals (meV) between two molecules in the face-to-face stacking arrangement for [Pt(dmdt)$_2$] and [Ni(tmdt)$_2$], as a function of a shift along the long molecular axis (*L*) and a shift along the short molecular axis (*s*). Interplanar distance *h* is fixed to 3.5 Å. The symbols p and c indicate transfer integrals, $t_{\text{pH-H}}$, $t_{\text{pL-L}}$, $t_{\text{cH-H}}$, and $t_{\text{cL-L}}$.

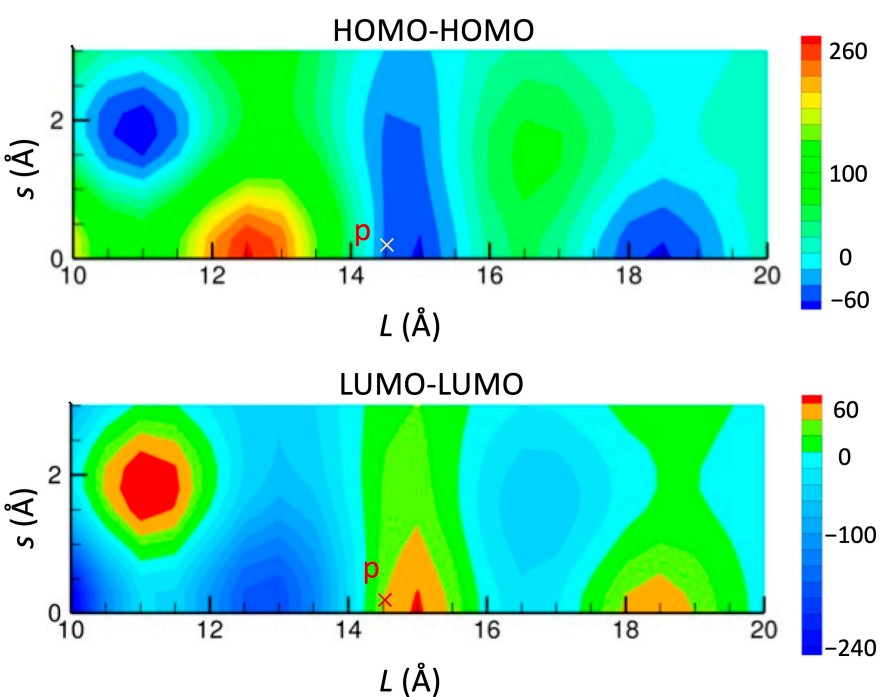

**Figure 16.** Map of HOMO–HOMO and LUMO–LUMO transfer integrals (meV) between two molecules in the face-to-face stacking arrangement for [Ni(btdt)$_2$], as a function of a shift along the long molecular axis (*L*) and a shift along the short molecular axis (*s*). Interplanar distance *h* is fixed to 3.0 Å. The symbol p indicates transfer integrals $t_{pH-H}$ and $t_{pL-L}$ (we do not consider $t_{cH-H}$, and $t_{cL-L}$ that are much smaller due to the longer interplanar distance).

### 4.2. Spin Ordering in [Ni(btdt)$_2$]: Antiferromagnetic HSE06 Calculations

To consider the AFM order, the magnetic unit cell needs to contain a minimum of two [Ni(btdt)$_2$] molecules, unlike the single molecule found in the unit cell of the experimental structure. To construct a magnetic cell, the original lattice vector of the crystal structure optimized using the vdW-DF2 functional is transformed using a rotation matrix consisting of (1, –1, 0), (1, 1, 0), and (0, 0, 1), with an origin shift of (0.5, 0.5, 0). The resulting lattice parameters are *a* = 6.98, *b* = 13.38, *c* = 14.46 Å, *α* = 110.43, *β* = 107.53, and *γ* = 69.32°.

We have found that a range-separated hybrid functional developed by Heyd, Scuseria, and Ernzerhof (HSE) is effective in describing the electronic states of molecular charge order systems [37,38]. The hybrid functional method calculates the exchange-correlation energy by combining the exact Fock exchange with the exchange energy functional from GGA.

As shown in Figure 17a, a finite band gap of 0.034 eV is obtained from spin-polarized HSE06 calculations. In the HSE calculations, the magnetic moments at the Ni site and (btdt)$_2$ ligands are ±0.05 and ±0.18 $\mu_B$, respectively. Thus, the total magnetic moment per [Ni(btdt)$_2$] molecule is 0.23 $\mu_B$. The magnetic moment of the ligand is determined by the summation of the magnetic moments of the C and S atoms belonging to the ligands. In contrast, the GGA calculations yield a magnetic moment of ±0.02 $\mu_B$ at the Ni site and ±0.06 $\mu_B$ for the ligands. A small band gap of approximately 0.01 eV is opened within the spin-polarized GGA, as plotted in Figure 17b.

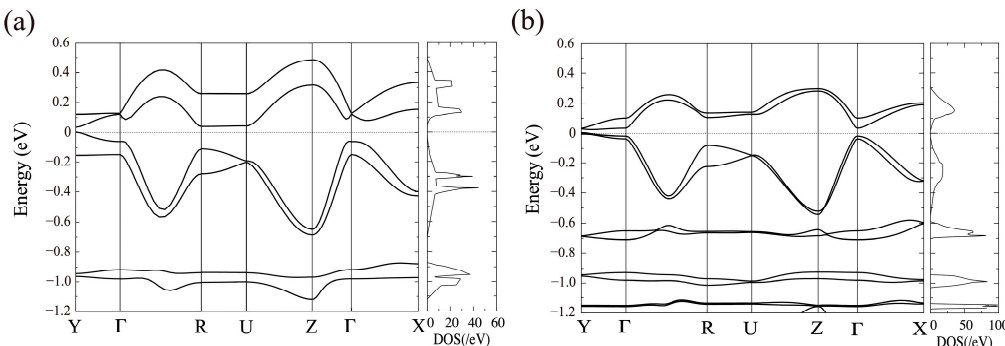

**Figure 17.** Band dispersion and total DOS for the AFM state in [Ni(btdt)$_2$] calculated with (**a**) HSE06 and (**b**) GGA-PBE functionals. The zero energy at the longitudinal axis is set at the top of the valence bands. Y = (0,1/2,0), Γ = (0,0,0), R = (1/2,1/2,1/2), U = (1/2,0,1/2), Z = (0,0,1/2), and X = (1/2,0,0), in units of the reciprocal lattice vectors.

Next, we discuss the AFM pattern of spin ordering. Figure 18 illustrates the spin densities in the AFM state of [Ni(btdt)$_2$]. As shown in Figure 18a, the spins alternate in a pattern along the *b* axis of the magnetic unit cell. The spin densities are mainly localized on the four S atoms closest to the Ni atom, as depicted in Figure 18b. The presence of the $d_{xz}$ orbital of the Ni atom indicates that the LUMO of the [Ni(btdt)$_2$] molecule contributes to the spin densities (see supporting materials of Ref. [13]). We note that these spin density and DOS analyses have been widely used in various other studies that describe the electronic structure of materials [39,40].

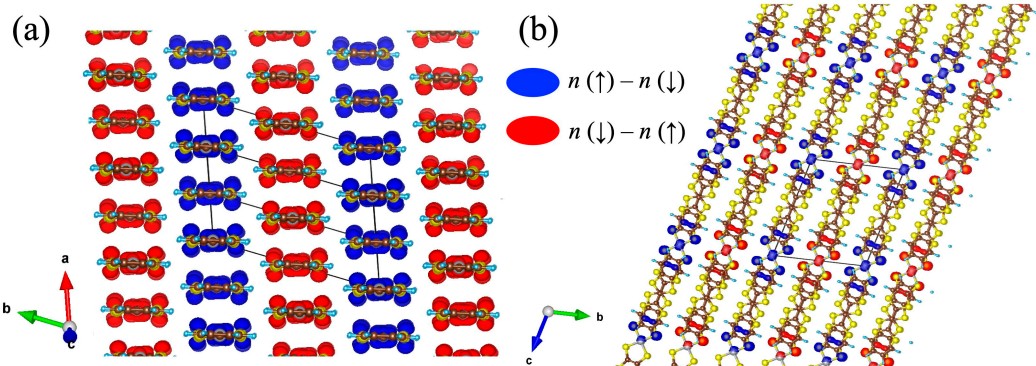

**Figure 18.** Contour plots of the spin densities of an AFM state in [Ni(btdt)$_2$]. (**a**) *ab* and (**b**) *bc* planes. The isosurface level of the spin density is set as 0.00043. The figures are plotted using VESTA software version 3 [41]. $n$ (↑) and $n$ (↓) are the spin-up and spin-down densities calculated with a HSE06 functional.

In the experiments, the magnetic susceptibility above 25 K follows the Curie-Weiss law, with the constants $C$ (0.52 emu K mol$^{-1}$) and $\theta$ (−28.63 K), indicating that intermolecular antiferromagnetic coupling is the dominant factor [13]. We obtain a stable AFM state through the spin-polarized HSE calculations using the structure optimized by the vdW-DF2 functional. On the other hand, using the experimental structure [13], we can stabilize an AFM state with a different ordering pattern from Figure 18, which was found to be metallic. Although the HSE functional includes a portion of the exact exchange, it does not account for the correlation effect. In the future, a semiconducting band structure can be achieved using methods that consider the correlation effect, such as GW approximation and dynamical mean field theory calculations.

## 5. Conclusions

Both the tight-binding model and the DFT calculation have indicated the possibility of Dirac cone formation in two single-component molecular conductors derived from metal complexes with extended tetrathiafulvalenedithiolate ligands, [Ni(tmdt)$_2$] and [Ni(btdt)$_2$]. Despite various band calculations performed on these systems, the formation of Dirac cones has never been reported before. This is because the close relationship between single-component molecular conductors and the Dirac electron system has not been recognized. Our proposed mechanism suggests that the stretcher bond type molecular arrangements favor the formation of Dirac cones, providing an important clue for the development of new Dirac electron systems. The manifestation of the Dirac electron system depends primarily on the proximity of the Dirac point to the Fermi level. Furthermore, the study of [Ni(btdt)$_2$] has revealed the importance of antiferromagnetic spin order effects. As observed in [Ni(btdt)$_2$], the results obtained from the tight-binding model and DFT calculations do not always agree. Although resolving this discrepancy remains a future challenge, this work has demonstrated that combining the strengths of both methods is a powerful tool for the design of functional molecular materials.

**Author Contributions:** Conceptualization, R.K. and T.T.; methodology, R.K. and T.T.; formal analysis, R.K. and T.T.; investigation, R.K. and T.T.; writing—original draft preparation, R.K. and T.T.; writing—review and editing, R.K. All authors have read and agreed to the published version of the manuscript.

**Funding:** This research was funded by a Grant-in-Aid for Scientific Research (JP19K21860) from the Japan Society for the Promotion of Science (JSPS) and JST, CREST Grant Number JPMJCR2094, Japan. This work was performed under the GIMRT Program of the Institute for Materials Research (IMR), Tohoku University (Proposal No. 202012-RDKGE-0034 and 202212-RDKGE-0048).

**Institutional Review Board Statement:** Not applicable.

**Informed Consent Statement:** Not applicable.

**Data Availability Statement:** The data are available from corresponding authors.

**Acknowledgments:** The study's computations were mainly conducted using the computer facilities of MASAMUNE at IMR, Tohoku University, Japan.

**Conflicts of Interest:** The authors declare no conflict of interest.

## Appendix A. Crystal Data of [Ni(btdt)$_2$] Optimized with the vdw-df2-b86r Functional

Space group: $P\bar{1}$
Lattice constants: $a_o = 6.36$, $b_o = 8.57$, $c_o = 14.46$ Å, $\alpha_o = 113.28$, $\beta_o = 78.34$, $\gamma_o = 126.73°$
Fractional atomic coordinates:

| Atom | $x$ | $y$ | $z$ |
|------|------|------|------|
| Ni | 0.0000 | 0.0000 | 0.0000 |
| S1 | 0.2206 | −0.0388 | 0.0741 |
| S2 | −0.1432 | 0.1154 | 0.1400 |
| S3 | 0.3365 | 0.0884 | 0.2961 |
| S4 | −0.0134 | 0.2300 | 0.3580 |
| S5 | 0.4602 | 0.2102 | 0.5300 |
| S6 | 0.1176 | 0.3598 | 0.5927 |
| C1 | 0.1804 | 0.0637 | 0.1994 |
| C2 | 0.5499 | 0.3109 | 0.7322 |
| C3 | 0.0192 | 0.1324 | 0.2288 |
| C4 | 0.1948 | 0.1925 | 0.3948 |
| C5 | 0.2495 | 0.2466 | 0.4943 |
| C6 | 0.4262 | 0.3049 | 0.6600 |

| Atom | $x$ | $y$ | $z$ |
|------|-----|-----|-----|
| C7 | 0.2691 | 0.3799 | 0.6900 |
| C8 | 0.2393 | 0.4651 | 0.7923 |
| C9 | 0.3672 | 0.4743 | 0.8641 |
| C10 | 0.5199 | 0.3961 | 0.8342 |
| H1 | 0.1208 | 0.5274 | 0.8155 |
| H2 | 0.3477 | 0.5446 | 0.9444 |
| H3 | 0.6189 | 0.4020 | 0.8906 |
| H4 | 0.6678 | 0.2480 | 0.7085 |

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
