# Peer review of "Dirac Cone Formation in Single-Component Molecular Conductors Based on Metal Dithiolene Complexes"

_magnetochemistry, doi:10.3390/magnetochemistry9070174_

Round 1

Reviewer 1 Report

In this work, Kato et al. employed a computational approach to provide information of Dirac cone formation in single-component molecular conductors based on metal dithiolene complexes. The Dirac electron system possesses a unique property known as the Dirac cone, which can provide a path to the next-generation quantum computing, superconductors and desktop relativistic mechanics and technology, so the work fits the current trends in chemistry and technology well. The work presents very interesting research results. It is also well written. The calculations have been done carefully and no obvious errors or omissions could be detected by referee. My opinion is that the topic is of interest to the readership of Magnetochemistry so the manuscript is publishable. However, I have minor edit comments:

1. In tables 1, 2 and 3 and throughout the text, the notation H-H, L-L for HOMO-HOMO and LUMO-LUMO intermolecular transfer integral should be used instead of just H and L, respectively.

2. There is an unnecessary line in equation (1).

3. On page 8, the numbers of figures in the text are wrong. It should be on: line 176 Figure 8a instead of 7a, line 183 Figure 8b instead of 7b, line 187 remove Figure 8a, line 190 Figure 9a instead of 8a.

4. The text lacks a description of Figure 11.

Reviewer 2 Report

The manuscript presents an intriguing study on Dirac cone formation in single-component molecular conductors, specifically using two nickel complexes with extended tetrathiafulvalenedithiolate ligands. The work is comprehensive, involving both theoretical modeling and first-principle density functional theory (DFT) calculations, and provides important insights into the formation of Dirac cones and the impact of different molecular arrangements. The use of DFT calculations is well justified and provides compelling evidence for the arguments put forth in the study. Therefore, I think that this work can be accepted in this journal after some minor revisions that may help to improve this version:

1. Many researchers use different display software such as Gaussview, vesta, material studio, etc. This can cause misinterpretation of the color of the element. Therefore, I recommend adding color annotations corresponding to the elements in the description of Figure 1, Figure 2, and Figure 8. Yellow S, gray C, and other atoms.

2. The manuscript could benefit from further clarification in the 'methods' section. For instance, the reasons for choosing specific k-point samplings, cut-off energies, and functionals could be briefly discussed to provide better understanding for readers less familiar with these techniques.

3. The author calculates the [Ni(tmdt)2] and [Ni(btdt)2] in this paper. I am curious about whether the author takes the influence of spin multiplicity into consideration in DFT calculations. The Ni atoms tend to have high spin multiplicity or low spin state?

4. The methods used in the article, such as spin density, dos orbital analysis, have been widely used in many other works describing the electronic structure, which are often used in the calculation of structures, catalysis, and batteries, such as Nanoscale (10.1039/d2nr06665c), Small (10.1002/smll.202206750), ACS Appl. Mater. Interfaces (10.1021/acsami.2c14041), Adv. Energy Mater. (10.1002/aenm.202300790) could be mentioned.

Minor editing of English language required
